# An Azure ACES Early Warning System for Air Quality Index Deteriorating

**DOI:** 10.3390/ijerph16234679

**Published:** 2019-11-24

**Authors:** Dong-Her Shih, Ting-Wei Wu, Wen-Xuan Liu, Po-Yuan Shih

**Affiliations:** 1Department of Information Management, National Yunlin University of Science and Technology, 123, Section 3, University Road, Douliu 640, Taiwan; D10423005@yuntech.edu.tw (T.-W.W.); tso2984986@gmail.com (W.-X.L.); 2Department of Finance, National Yunlin University of Science and Technology, 123, Section 3, University Road, Douliu 640, Taiwan; D10424003@yuntech.edu.tw

**Keywords:** air pollution, machine learning, AQI, Azure, cloud computing

## Abstract

With the development of industrialization and urbanization, air pollution in many countries has become more serious and has affected people’s health. The air quality has been continuously concerned by environmental managers and the public. Therefore, accurate air quality deterioration warning system can avoid health hazards. In this study, an air quality index (AQI) warning system based on Azure cloud computing platform is proposed. The prediction model is based on DFR (Decision Forest Regression), NNR (Neural Network Regression), and LR (Linear Regression) machine learning algorithms. The best algorithm was selected to calculate the 6 pollutants required for the AQI calculation of the air quality monitoring in real time. The experimental results show that the LR algorithm has the best performance, and the method of this study has a good prediction on the AQI index warning for the next one to three hours. Based on the ACES system proposed, it is hoped that it can prevent personal health hazards and help to reduce medical costs in public.

## 1. Introduction

The degree of air pollution has risen in recent years and has a direct impact on urban pollution and people’s health, especially in developing and industrial countries where there is no or only minimal air quality management [1]. Daily predictions of pollutant concentrations in the atmosphere are very important for regulatory planning. When harmful events are predicted, information is provided to the public and social activities are restricted in advance. If early and effective early warning systems are established, casualties and negative impacts on human beings can be greatly reduced [2]. Air pollution early warning system is a very useful tool for avoiding adverse health effects and formulating effective prevention programs, but the development of a strong early warning system is very challenging, but also necessary [3].

In 2017, Taiwan set a new air quality index (AQI) with reference to American standards. It not only integrates the old PSI (Pollutant Standards Index) and PM_2.5_, which are not easy to interpret, but also is the most widely used index in many different countries in the world [4]. It can precisely remind people of self-protection. According to research, air pollutant exposure is strongly associated with asthma and lung diseases [5,6]. The study published in the American Heart Association Journal Hypertension concludes that short-term exposure to SO_2_, PM_2.5_ and PM_10_ increases the incidence of hypertension. According to the World Health Organization (WHO), 92% of the world’s population lives in areas where air quality levels exceed their organizational limits, and 3 million deaths per year for human health are related to outdoor air pollution. In 2017, about 1.7 million children under the age of 5 died each year due to environmental health problems such as air pollution, accounting for more than a quarter of the total number of children dying in this age group.

The harm of air pollution to people’s health also results in huge medical expenditure of diseases derived from it every year. According to the research and estimation of European and American countries, every year when the life span of individuals is reduced, the society must pay NT$2 million in labor loss, care, and medical expenditure and other costs. According to a 2010 Rand Medical and Health Research Report on Air Pollution in California, the number of patients hospitalized in California during 2005–2007 was as high as 30,000, resulting in medical costs of up to $190.3 million [7]. In China, where air pollution is very serious, the high mortality rate and health care costs caused by pollution are about $300 billion a year, and as many as 500,000 people die prematurely each year because of air pollution. The Organization for Economic Cooperation and Development (OECD) published a report on the economic consequences of air pollution in 2016. Air pollution is causing delays in work, reduced agricultural production and increased medical costs. health care expenditures have increased from $210 in 2015 to $176 billion, and misemployment losses from related diseases have increased from $1.2 billion to $3.7 billion. At the same time, air pollution will kill 6 million to 9 million people worldwide every year.

There are many state-of-the-art studies on air quality prediction such as Singh et al. [8] in spatial deterministic and Zhou et al. [9] in statistical forecasting. However, most of studies focus on the concentration prediction of PM_2.5_, which is different from the AQI predicted in this study. In the newer AQI-related air quality prediction research, Wang et al. [10] solved the factors that caused the prediction difficulties such as randomness, instability and irregularity in AQI research by using two-phase decomposition technology, and used the Extreme Learning Machine (ELM) to predict AQI. The proposed hybrid model based on two-phase decomposition technique and applicable to AQI prediction has obviously higher prediction accuracy than other models. Zhu et al. [11] have designed two mixed models for regional AQI index to carry out numerical prediction, and solved the shortcomings of using single model to grasp information comprehensively from the index, and improved the prediction accuracy with new effective technology. Chen et al. [12] have developed a prediction model based on neural network, which combines social media with monitoring sensors, and uses AQI related values as input variables to predict health hazards caused by smoke. This prediction method can provide decision-making information for health hazard management through early warning and other functions. In this study, data are collected from the Taiwan air quality monitoring network, which provides information on the air monitoring stations set up by the environmental protection department of the Executive Yuan.

In the past few years, machine learning algorithms have been widely used to detect potential patterns in various data streams and obtain predictive results [13,14]. However, with the change of data characteristics, scalable machine learning has become a necessary solution. The basic concept of scalable machine learning is to disperse computing to the cloud to accelerate the process of modeling [15]. With the increasing amount of data, the speed of data storage and reading, and more and more different types and sources of data, these problems can be solved by utilizing the advantages of infrastructure services such as cloud platform, and by designing prediction models with machine learning module [16].

This study will be completed on Azure cloud computing platform using cloud services, according to the characteristics of air quality monitoring data, with Microsoft Azure Machine Learning service. The AQI deterioration on-line warning alert is carried out by capturing the real-time air quality monitoring data updated hourly by the government. Thus, this study collects air quality monitoring data from January 2016 to May 2018 in Taiwan, establishes the prediction model of AQI pollutant concentration and attempts to browse the AQI warning in next six hours, since the government only provides the forecast information at least one day later. It is hoped that this study can help public to avoid approaching the areas with serious air pollution, and to reduce the health hazards to individuals.

## 2. Literature Review

### 2.1. Impact of Air Pollution

According to the WEO (World Energy Outlook) report of IEA (International Energy Agency), air pollution has become a major public health crisis. Nearly 6.5 million people around the world have died of poor air quality, making air pollution the fourth leading cause of human death in the world, and affecting the environment, economy, and food safety [17]. Air pollution is mainly caused by a large number of human energy production and use. The WHO report also points out that most of the deaths and diseases caused by air pollution are related to PM_2.5_, i.e., particulate matter with a diameter less than 2.5 micrometers. Among them, carbon black, also known as short-term climate pollutant (SLCP), is the main component of PM_2.5_, which is harmful to human health, mostly from diesel vehicles, diesel engines, and so on [18]. Biomass Boiler and Waste Incineration. Another short-term climate pollutant, ozone, is a mixture of pollutants emitted from urban or nearby rural areas. Therefore, the burning of biomass and fossil fuels, along with people’s economic activities and the energy demand of many growing cities in the world, makes poor air quality a serious urban problem.

### 2.2. Air Quality Index AQI

There are many different standards for judging air pollution quality, and there will be some differences in the degree of air pollution judged under different standards. In 2017, Taiwan adopted AQI (Air Quality Index) as the formal criterion, so that people can have more simple and clear air quality information as the criterion for judging. Comparing the difference between AQI and PM_2.5_ index, grading color is added to the classification of low concentration, which can make AQI, even in the condition of ordinary air quality, more clearly understand the influence degree of air pollution at present, and keep the concentration (35 µg/m^3^) of the warning focus of the original PM_2.5_ index and give cautious suggestions. Air quality index AQI, a new air quality index set by EPA of Taiwan Executive authorities, refers to American standards. Compared with the old ones, AQI adds the moving average value of PM_2.5_ pollutants and ozone (O_3_) for 8 h to the sub-index of AQI judgment, and becomes the judgment basis of the latest air quality standard in Taiwan.

The AQI value ranges from 0 to 500 and is divided into six different pollution levels by six colors. The calculation of AQI is based on the concentration values of ozone (O_3_), fine suspended particulate matter (PM_2.5_), suspended particulate matter (PM_10_), carbon monoxide (CO), sulfur dioxide (SO_2_) and nitrogen dioxide (NO_2_). With its impact on human health, the individualized air quality index (IAQI) of different pollutants was calculated by Formula (1). Then the maximum of each indices was selected by Formula (2) to determine the final air quality index (AQI). Detailed formulas, symbolic descriptions and AQI indicators calculation comparison table shown in Table 1:(1)$$IAQI_{p}=\frac{IAQI_{Hi}-IAQI_{Lo}}{BP_{Hi}-BP_{Lo}}\left(C_{p}-BP_{Lo}\right)+IAQI_{Lo}$$
(2)$$AQI=max\left\{IAQI_{1},IAQI_{2},IAQI_{3},\cdots ,IAQI_{n}\right\}$$

### 2.3. Relevant Research on Existing Air Pollution and AQI

#### 2.3.1. Study on the Impact of Air Pollution

Pan et al. [19] used the Gauss distribution model to analyze the impact of traffic flow and regional carbon monoxide concentration. Finally, it was confirmed that there was a significant relationship between traffic flow and regional carbon monoxide concentration. Statistical analysis was used to study the effects of air pollution and suicide in Tokyo from 2001 to 2011, and positive results were obtained [20], Hjortebjerg et al. [21] have studied the effects of maternal exposure to air pollution and traffic noise on the number of births of newborns. Deng et al. [22] assessed the association between outdoor air pollution and allergic rhinitis in children, Lee et al. [23] and others have studied the effects of air pollution on Parkinson’s disease, Lichter et al. [24] found that air pollution was negatively correlated with the performance of German football players. Kingsley et al. [25] explored the relationship between air pollution in pregnant women’s living areas and fetal development according to their geographical location, assessed the levels of pollutants in women and infants, and investigated the results through linear regression.

Research by literature review methods, Vizcaino et al. [26] systematically analyze the adverse effects of outdoor air pollution on human infertility, Chen et al. [27] use the literature to outline the effect of UFP (ultrafine particles) on adverse health effects. Santibáñez-Andrade et al. [28] also used a literature review to explore the relationship between air pollution and lung cancer, and found that air pollution in addition to smoking also has a certain risk for lung cancer. For the time series, Ma et al. [29] analyzed the relationship between patients hospitalized for cardiovascular disease in Beijing and air pollution, and found significant effects with men older than 65 years. Li et al. [30] tried to explore the impact of these variables on PM_2.5_ by using PM_10_, weather variables and spatial effects to estimate the temporal and spatial concentration of historical PM_2.5_. The results show that these variables are the most important in autocorrelation prediction.

#### 2.3.2. Research on AQI and Other Air Pollution

There are many studies aimed at predicting air quality index. Machine learning is the most common method used in predictive research. Perez and Gramsch [31] used neural networks to predict PM_2.5_ hourly concentration in Chile’s capital. Particularly, some events that cause concentration rise at night, such as traffic flow, were added as predictive variables. Their model can predict the concentration of PM_2.5_ in the next 24 h, and successfully warn the time when the concentration exceeds the standard from night to midnight. Zhan et al. [32] Established a continuous learning model for predicting daily PM_2.5_ concentration in China. In addition to its superior predictive performance, it can also deal with missing values, which can be used to assess the impact of acute human health. Wang et al. [10] used two-phase decomposition technology to improve the difficulty of AQI prediction with Extreme Learning Machine (ELM). Chen et al. [12] used the combination of social media and monitoring sensors to predict smoke health hazards by using AQI index as an input variable, Shaban et al. [33] also carried out systematic monitoring and prediction for the three most harmful gases released by WHO. Detailed air pollution-related research can be shown in Table 2.

## 3. Methodology

### 3.1. System Architecture

The overall ACES (Azure Computing and Evaluate Services) system framework is built on Microsoft Azure Cloud, which uses App Service and Machine Learning to predict the deterioration of AQI index on-line and send warning messages to users. It is composed of different databases, and six modules. First, the data collection and pre-processing module stores and backs up the data after it’s collected. Then the Prediction Model Constructing and Applying Module reads the air quality data from the database and performs the prediction of the air quality index data. The results are stored and backed up again and transmitted to the Decision Module for the user with the warning message. If the predicted results compared with the AQI standard and exceed the standard values, the Early Warning Alert Module will be given the instructions to transmit warning messages to users, Finally, the system users can clearly understand the current AQI distribution by browsing the visualization map generated by data visualization module. The system architecture diagram and modules are shown in Figure 1.

#### 3.1.1. Data Collection and Preprocessing Module

Data Collection and Preprocessing Module is the first step of ACES system. First, two instant mechanisms called Pollutants Real-Time Data and PM_2.5_ Real-Time Data, contained in Time Module, the corresponding data collection function models “Pollutants Data Collection Model” and “PM_2.5_ Data Collection Model” are used to collect real-time data. It will request the Web of Taiwan Air Quality Monitoring Network to obtain the data.

Next, the original data is stored in the database and transmitted to the data preprocessing module for the data pre-processing. The first step is data cleaning, and the second step is to convert all data into the content needed for the early warning system, the last step is to integrate the data captured and processed from two different data collection function models, and then compare and merge them into the final required data form and store them in the Azure cloud, the processing module architecture is shown in Figure 2.

#### 3.1.2. Prediction Model Constructing and Application Module

Prediction Model Constructing and Applying Module will be divided into two parts: Firstly, the historical air quality monitoring data will be input into Preprocessing Process in Prediction Model Constructing Module, Training Data and Testing Data are input into Training and Testing processes respectively for training and testing of model building. In the model training phase, the training data will be iterated many times by the regression-type machine learning algorithms provided by Azure Machine Learning service to complete individual model training. In the test phase, the test data are input into individual training models, and the output results are compared with the actual values.

Next, in the Prediction Model Applying Module, the pre-processed data is obtained from the database and then processed by the feature engineering step to produce the data required for the prediction, finally input the best prediction model evaluated in the Prediction Model Constructing Module to predict the concentration of air pollutants and generated the predicted value into the database. The detailed module operation process described in this section is shown in Figure 3.

#### 3.1.3. Decision Module

The function of decision module is that after receiving the air pollutant concentration prediction value, the AQI calculation formula is used to calculate the side-index value of each pollutant, and then the highest value of the side-index value is selected as the real-time AQI value and compared with the level Table. If the deterioration of AQI exceeds the general standard, the Early Warning Alert Module will be given an early Warning Alert Decision function. The decision execution steps of this module are shown in Figure 4 and Figure 5.

#### 3.1.4. Early Warning Alert Module

Early Warning Alert Module only operates when it receives instructions from Decision Module to send warnings. After receiving high level data and instructions, it checks the area where AQI exceeds the standard with the area where all users in the database are located and send alert to users in relevant areas. the process of sending warning messages by this module is shown in Figure 6.

### 3.2. System Environment

The ACES early warning system of this study is built using Visual Studio 2017 version and Microsoft Azure cloud platform. The Azure uses the level of effectiveness of functions as shown in Table 3 below.

#### 3.2.1. Establishment and Deployment of Azure Environments

ACES early warning system will use four kinds of services in Azure, namely App Service, SQL Database, Machine Learning Studio and Storage. Firstly, App Service will be established to deploy the completed system project to the cloud. Then, the database of system data storage will be established with the function of SQL Database. Then, Learning Machine Studio will be established. Functions and Storage can be completed.

#### 3.2.2. Establishment and Deployment of Prediction Model

The Machine Learning prediction model used in the operation of the system is constructed by using Machine Learning Studio. First, the data set for training is uploaded. Then the experiments are established for each model. The prediction model can be built according to the requirements in each experiment. the prediction model is deployed to the network using Web Service, detailed picture shown in Figure 7.

## 4. Experiment

### 4.1. Procedure

The experimental procedure is divided into three stages: model training, testing and prediction. And, all the data set used and their duration are shown in Figure 8. The goal of model training is to find the best window size in training, that is, by using historical data set {*X*(t − n), *X*(t − n + 1), …, *X*(t)} what is the best n in forecasting *Y*(t + 1). Then, in model testing stage, three machine learning algorithms are tested to find the best algorithm in prediction for next prediction stage. A detailed flow chart of experiment is shown in Figure 9.

#### 4.1.1. Model Training

Due to the limitations Azure Machine Learning Studio, only per hour model can be established for each pollutant. AQI index came from six pollutants, SO_2_, CO, O_3_, PM_10_, PM_2.5_, NO_2_, in order to predict next six hours, a total of 36 separate prediction models are generated. The input data of model training is adjusted by “Time Series” method, which has been shown in Shaban et al. [33]. For example, the output value of SO_2_(t + 1) can be predicted by {SO_2_(t)}, or {SO_2_(t), SO_2_(t − 1)}, or {SO_2_(t), SO_2_(t − 1), SO_2_(t − 2)}, …, etc. base on window size 1, 2, 3, … etc. as shown in Figure 10. After testing from training data of SO_2_, it is shown that if window size is set to be 4 since it has the best performance in all measure than other window size as shown in Table 4. Therefore, all the model proposed in next section are all base on this result which is the output value of SO_2_(t + 1) is predicted by the input set of {SO_2_(t), SO_2_(t − 1), SO_2_(t − 2), SO_2_(t − 3)}. The predicted value of SO_2_(t + 1) is defined as SO_2_*y*(t + 1).

When it comes to predicting the next six hours (t + 1, t + 2, ..., t + 6), the predicted value SO_2_*y*(t + 1) of SO_2_(t + 1) is added to the prediction of SO_2_(t + 2) whose input set is {SO_2_*y*(t + 1), SO_2_(t), SO_2_(t − 1), SO_2_(t − 2), SO_2_(t − 3)} and so on. Finally, the prediction value of SO_2_(t + 6) is came from the prediction of the input set of {SO_2_*y*(t + 5), SO_2_*y*(t + 4), SO_2_*y*(t + 3), SO_2_*y*(t + 2), SO_2_*y*(t + 1), SO_2_(t), SO_2_(t − 1), SO_2_(t − 2), SO_2_(t − 3)}. A generic representation of model training is shown in Figure 11. Note that, AQI index has six pollutants and each pollutant use 15 variables to predict which will show in Section 4.2.

#### 4.1.2. Model Prediction

Because AQI index use 6 pollutants and pollutants at time t + 1 can be predicted with values at time t − 3, t − 2, t − 1 and t. Burgos et al. [38] substitute the future real values with the values predicted by their study, and then complete all stages of the prediction. This is more like an incremental learning in this prediction process. Five predicted vectors, *Y*(t + 1) to *Y*(t + 5), are replaced with *X_Y_*(t + 1) to *X_Y_*(t + 5) and the pollutant prediction process of the next one to six hours will be completed, and the results will be used for subsequent AQI calculation. The detailed generic model prediction process is shown in Figure 12. That is, the pollutant prediction process of the next one to six hours will be completed, and the results will be used for subsequent AQI calculation. The detailed model prediction process is shown in Figure 12 below.

### 4.2. Air Quality Index Data

This study collects air quality monitoring data from January 2016 to May 2018 in Taiwan for training and testing model, establishes the prediction model of AQI pollutant concentration, and obtains the latest air quality monitoring data through the system’s real-time data collection program for real-time prediction, supplemented by the relevant variables contained in the monitoring data that may affect the prediction results. As shown in Table 5, *y_i_*(t + 1),…,*y_i_*(t + 6) are the output variables where *I* = 1, 2, …, 6, *x_j_*(t − 3),…,*x_j_*(t) are the input variable where *j* = 1, 2, …, 15, and *x_yi_*(t + 1),…,*x_yi_*(t + 5) are the input variable of the next stage which are the predicted value of *y_i_*(t + 1),…,*y_i_*(t + 5) in the model.

### 4.3. Evaluation

After the establishment of the prediction model, the performance of each model and the predicted index results are compared. The corresponding evaluation indicators are used to evaluate each model. Therefore, the indicators of model evaluation and the air quality index evaluation will be explained in this section.

#### 4.3.1. Evaluation Indicators

Through the prediction models trained by three machine learning algorithms, we need to use appropriate model evaluation indicators to judge each model. We can verify the prediction accuracy errors between the predicted AQI values of the system and the actual AQI values published by the government afterwards, and select the best model with the highest performance. The following indicators are selected to evaluate the prediction model for the model of regression algorithm in this study:

(1) Mean Absolute Error (MAE)

The mean absolute error has the same unit as the original data. It can only be compared between models whose errors are measured in the same unit. It is used to measure how close the prediction is to the actual results. Its calculation formula is shown in Formula (3):(3)$$MAE=\frac{\sum_{i=1}^{n}\left|Predicted_{i}-Actual_{i}\right|}{n}$$

(2) Root Mean Squared Error (RMSE)

The root mean square error is a popular formula to measure the error rate of regression models, but only when the errors are compared between the models measured in the same unit, a single value of error in the aggregate model will be generated. By means of square difference, the measurement ignores the difference between over-prediction and under-prediction, and can be used to measure the difference between the predicted value and the actual value. The calculation formula is shown in (4):(4)$$RMSE=\sqrt{\frac{\sum_{i=1}^{n}{\left(Predicted_{i}-Actual_{i}\right)}^{2}}{n}}$$

(3) Coefficient of determination

Usually referred to as R^2^, this paper describes the proportion of mean square deviation of dependent variables explained by regression models, whose values range from 0 to 1. The calculation formulas are shown in Formulas (5)–(8):(5)$$R^{2}=\frac{SSR}{SST}=1-\frac{SSE}{SST}$$
(6)$$SST=\sum {\left(y-\bar{y}\right)}^{2}$$
(7)$$SSR=\sum {\left(y^{\prime }-\bar{y^{\prime }}\right)}^{2}$$
(8)$$SSE=\sum {\left(y-y^{\prime }\right)}^{2}$$

#### 4.3.2. Assessment Indicators of Air Quality Index

The evaluation of the predicted value of AQI pollutants predicted by the model is based on the comparison table of AQI pollutant concentration and instant by-index value promulgated by the Environmental Protection Department of Taiwan Executive. According to different AQI values, there are six colors representing different degrees, Comparing the evaluation criteria table, the detailed comparison table of AQI indicators is as follows with Table 6.

## 5. Experimental Results and Discussion

### 5.1. Data Collection and Processing

The data collected in this study are from air real time data of monitoring stations published by EPD of the Executive. After data pre-processing and deletion, 940,000 data were collected in 2016 and 2017, and 190,000 data were collected in January–May 2018. Therefore, the total number of historical datasets is about 1.14 million.

#### 5.1.1. Data Collection

The historical data from January 2016 to May 2018 were obtained from Excel files classified according to the year and month of each station published by EPD, the first-hand collected data were not entirely consistent with the needs of this study. Therefore, the results of data pre-processing are described in the next section. The original collection structure of historical and real-time data is shown in Table 7 and Table 8.

#### 5.1.2. Data Processing

Due to the large number of historical data files classified according to the year and month of each station, it is necessary to use a function to pre-process Excel files, through data cleaning, conversion and merging, and not all monitoring stations monitored the same items, among which the THC, NMHC, CH4, UVB, PH_RAIN and RAIN_COND fields are less than half of the total data, so the data fields are not equal. Then field was deleted to reduce 21 items to 15 items. Finally, in view of the lost value processing caused by the maintenance of the station equipment and other reasons, after the small sample interpolation method and the actual test of the deleted data, this study finds that the performance of deleting the data directly is higher, and the data processing of instant collection will be processed and transformed directly in the code during the operation of the system, and the data will be processed to meet the prediction.

### 5.2. Experimental Results and Performance

This section will describe and explain the training, testing and prediction results of the model respectively. This study was carried out in Douliu City, Taiwan. The trend breakdown and regression analysis produced in each stage of this chapter will take Douliu Monitoring Station as an example.

#### 5.2.1. Model Training

This study tested the performance of three supervised machine learning algorithms: Decision Forest Regression (DFR), Linear Regression (LR) and Neural Network Regression (NNR). According to the results of each performance evaluation index, the best model was selected and the machine learning algorithm used as the follow-up research was determined. Prediction model with Table 9 is the result of using data from August 2016 to December 2017 for model training and using data from January 2016 to July 2016 for testing and evaluating the first-hour performance of six pollutant algorithms. Most of the LRs have the best or the second-best performance under each performance index.

After the algorithm has determined and established a total of 36 prediction models for six pollutants, the data used in the training model are re-entered into the prediction model to try to understand the prediction performance of each model based on training data. The most important AQI numerical prediction R^2^ ranges from 0.897 of *Y*(t + 1) to 0.97 of *Y*(t + 6). The six pollutants also showed poor performance in *Y*(t + 1) as a whole, but the performance of each indicator increased obviously from *Y*(t + 2). Although the performance decreased gradually from *Y*(t + 2) to *Y*(t + 6), the change was not significant, perhaps the data itself was the same as the data set used in the training model. The overall performance of the detailed training phase data forecast is shown in Table 10.

#### 5.2.2. Model Testing

In this study, all the data from January 2018 to May 2018 were used to the test model. In this stage, all data were pre-processed and input into the pollutant prediction model established in the previous stage. The prediction performance of the test stage was through the overall performance Table, trend breakdown chart and regression analysis of Douliu City. The overall performance of the detailed test phase data prediction is shown in Table 11. The overall prediction results show that although the predicted performance of each pollutant is good at *Y*(t + 1), the performance indicators from *Y*(t + 2) to *Y*(t + 6) begin to decrease dramatically. Except that the R^2^ of AQI can keep at 0.683 at the lowest level, the R^2^ of other pollutants is lower than 0.5. Although the air quality warning standard can be maintained within the ideal standard range, other pollutant predicted as AQI calculations may need to be adjusted to make the calculated more accurate for *Y*(t + 2) to *Y*(t + 6). However, it indicates that the public can take into account that the next hour prediction in AQI index is the best in next six hour’s prediction and its average performance of R^2^ is over 0.981 (in Table 11) since it is approach to 1 which is the idea value of R^2^. AQI predicted results in Douliu City in May 2018) is also shown in Figure 13 for demonstration, its R^2^ is 0.9611 also approach to 1 too. From Table 11, we can also conclude that our proposed system has an excellent prediction in next two hours since its R^2^ are over 0.936 in model testing.

#### 5.2.3. Model Prediction

From 1 June 2018 to 30 June 2018, all the predicted results and actual values are analyzed and compared. From the overall performance of the actual predicted in third stage, it is also found that the performance of each pollutant at *Y*(t + 1) is well, but from *Y*(t + 2) to *Y*(t + 6) there is a gradual decline. The overall performance of the detailed prediction stage is shown in Table 12. AQI index remain the best performance in R^2^, it reaches 0.947, *Y*(t + 1) almost close to 1 in next hour prediction.

In the prediction stage, the trend break-line chart and regression analysis chart of Douliu City for the next hour in June 2018 is shown in Figure 14. Basically, most of the pollutant performance indicators and AQI are almost the same as May 2018 in Figure 13. Nevertheless, after close examine the performance in every month in 2018, the results are almost the same. It shows our proposed ACES system’s robustness.

### 5.3. Discussion

The method proposed in this study has a good performance in predicting the results of AQI in the first hour in the test and prediction stage, while the predictive performance of AQI in the fourth to sixth hours is relatively low. Compared to other pollutants, the performance of SO_2_ in the six pollutants is relatively not well. Although the predictive performance of the six pollutants is not the same, the AQI values calculated in the end all show good performance. A possible reason is the maximum AQI is selected as the representative. When the pollutant value is incorporated into the AQI formula, the influence of the prediction error of the pollutant value itself may be indirectly reduced by the calculation method of the formula.

The reason why the predictive performance of model *Y*(t + 1) to *Y*(t + 6) declines gradually may be that the variables characteristic used in the current model are not considered traffic, factories and other possible variables. Maybe that’s why the next hour prediction is good enough and decay as times go by.

Azure machine learning studio is still in developing progress, therefore, this study only chooses the matured three machine learning algorithm in prediction. Maybe by adding code from R and Python or use another matured machine learning algorithm in prediction will achieve a better result in the fourth to sixth hours’ AQI index prediction. The poor performance of the prediction model in the fourth to sixth hours may also be that the old data need to be retrained. This study did not carry out the model retraining. When the system is officially running, a threshold value of error standard can be set to check the prediction performance every time or regularly. If the error is greater than the set threshold, the latest data will be added to the model training data to retrain the model. A possible detailed model retraining process is shown in Figure 15.

## 6. Conclusions

For the achievements and contributions of this study, first of all, a set of air quality deterioration early warning system integrated by Azure services is proposed, and its cloud-based architecture has many advantages over the use of local servers, such as easy maintenance and management, providing a series of highly integrated and compatible functions, and easy expansion of efficiency. The experimental results show that the ACES system has good prediction results for the AQI index for the next one to three hours, and it also provides users with visual distribution map service of air pollution in Taiwan’s counties. Unaware of the shortcomings of future AQI predictions in hourly units, an information-based intervention to help people in advance or avoid approaching areas with serious air pollution will reduce personal health hazards and medical costs. Finally, comparing this study with some other related study, we find that although the prediction range of this study is relatively short, most of the studies seldom use cloud platform, and don’t have fully applied such as early warning and the visualization map. The study comparison is shown in Table 13.

In the future, institute has limited types of data related to air pollution and some sensitive data are difficult to obtain, so it cannot consider various factors that may affect air quality, such as urban traffic or factory exhaust. If there is an opportunity, more different kinds of data like open data can be added to improve the research. As for the model retraining mechanism, threshold issue and feature correlation analysis can also be further studied for future prospects.

## Figures and Tables

**Figure 1 ijerph-16-04679-f001:**
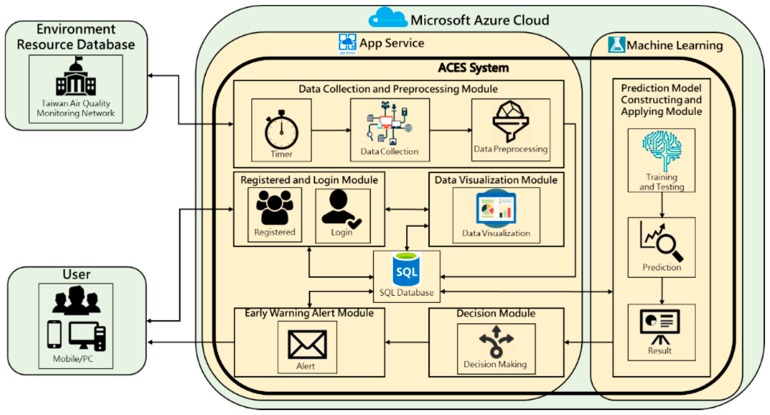
System Architecture Diagram.

**Figure 2 ijerph-16-04679-f002:**
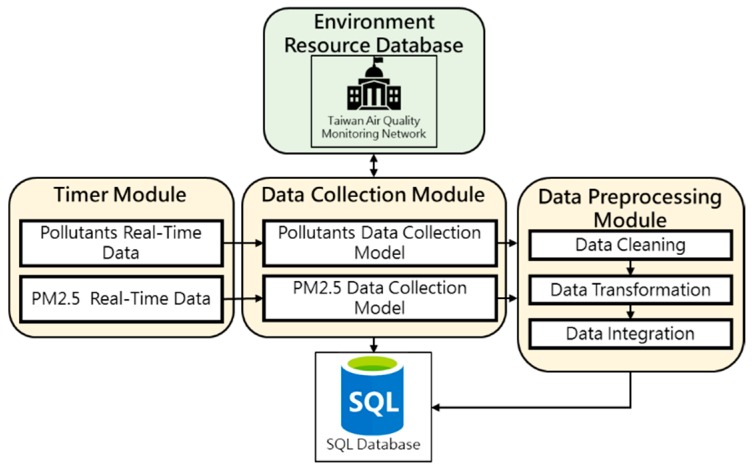
Data Collection and Preprocessing Module Diagram.

**Figure 3 ijerph-16-04679-f003:**
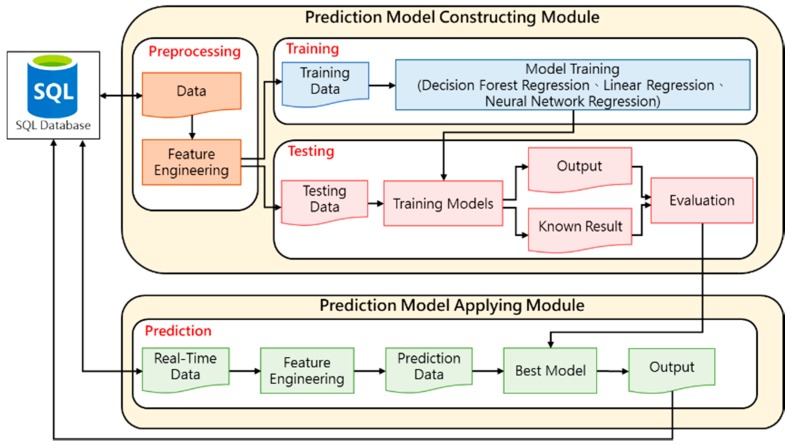
Prediction Model Constructing and Application Module Diagram.

**Figure 4 ijerph-16-04679-f004:**
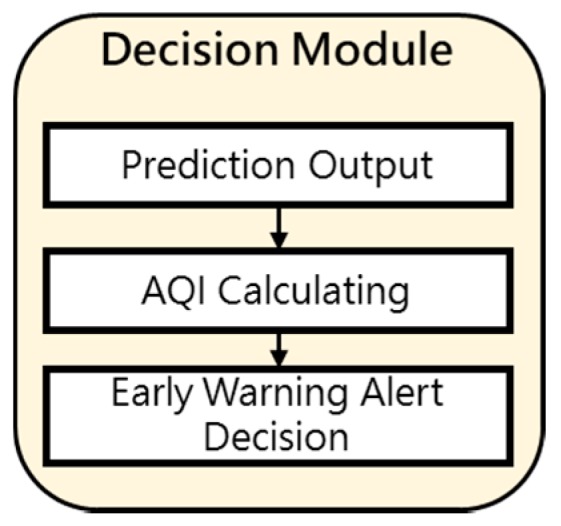
Decision Module Diagram.

**Figure 5 ijerph-16-04679-f005:**
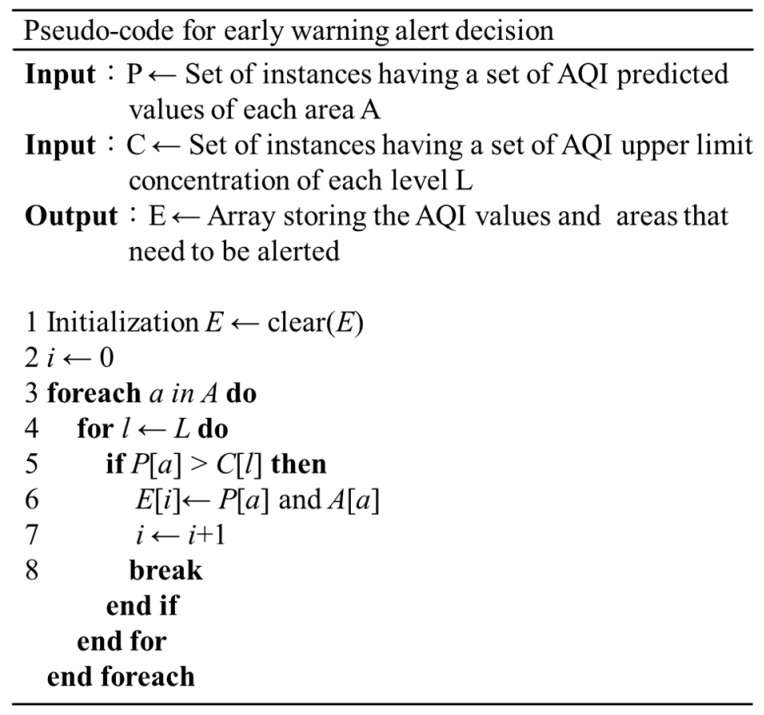
Algorithm for Early Warning Alert Decision.

**Figure 6 ijerph-16-04679-f006:**
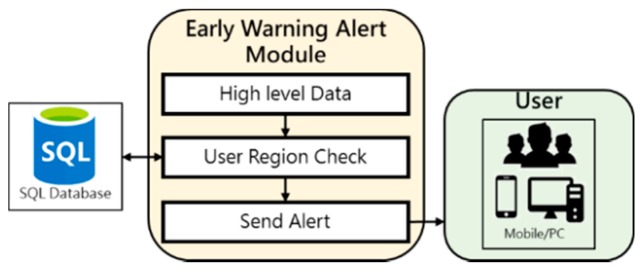
Early Warning Module Diagram.

**Figure 7 ijerph-16-04679-f007:**
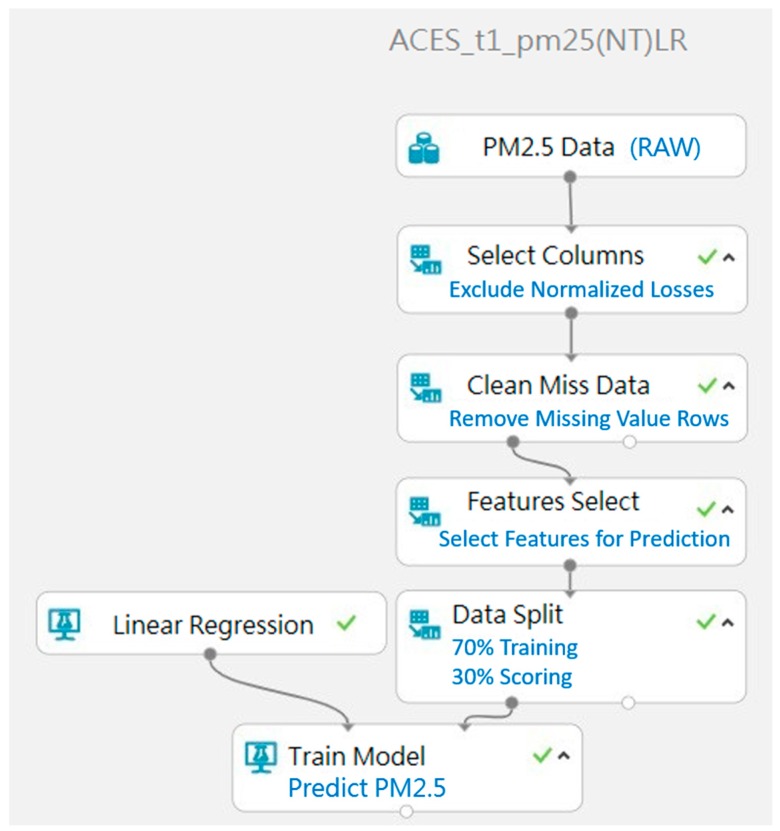
Prediction Model Establishment.

**Figure 8 ijerph-16-04679-f008:**
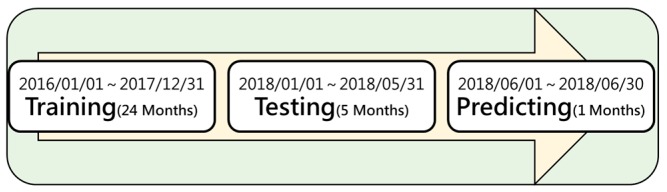
Time history chart of experimental data.

**Figure 9 ijerph-16-04679-f009:**
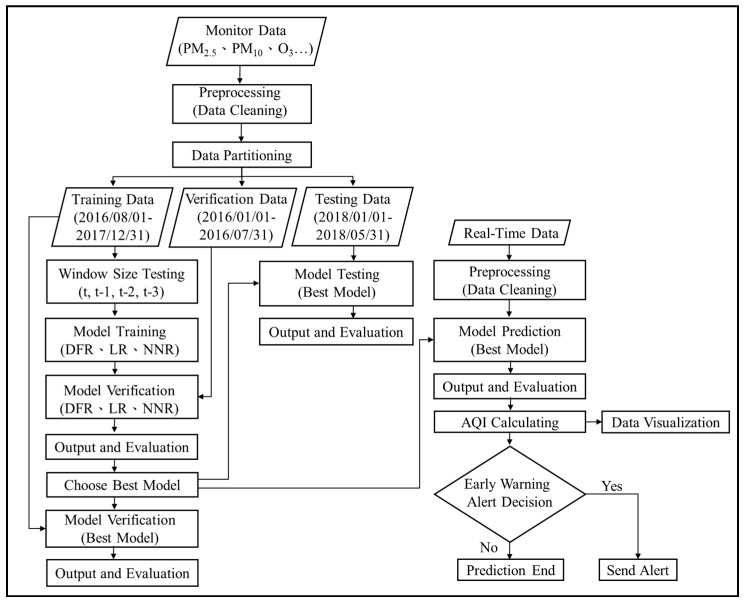
Overall experimental flow chart.

**Figure 10 ijerph-16-04679-f010:**
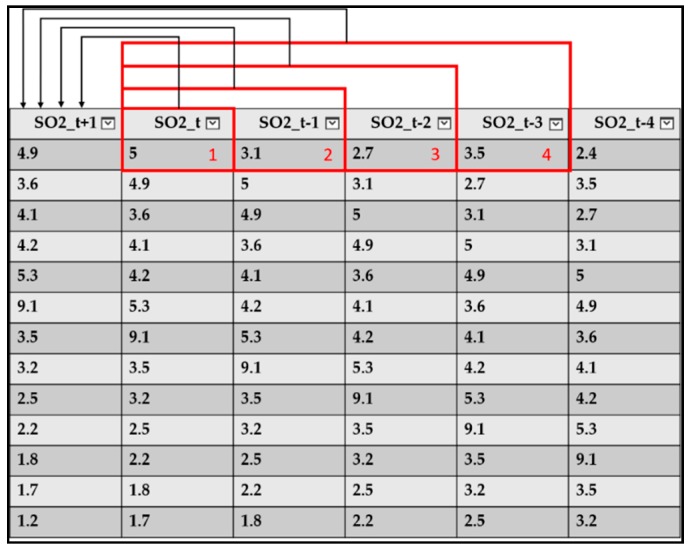
Window Size test schematic.

**Figure 11 ijerph-16-04679-f011:**
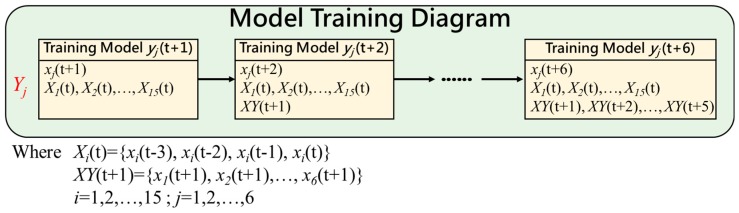
Genetic representation of model training.

**Figure 12 ijerph-16-04679-f012:**
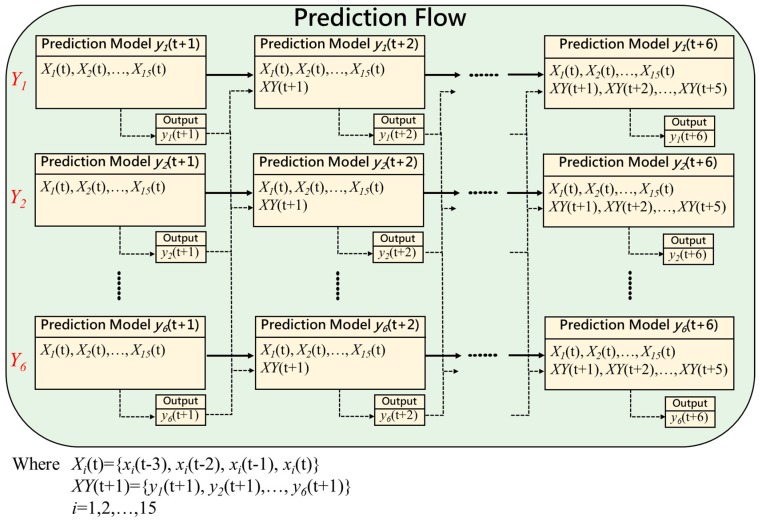
Flow chart of model prediction.

**Figure 13 ijerph-16-04679-f013:**
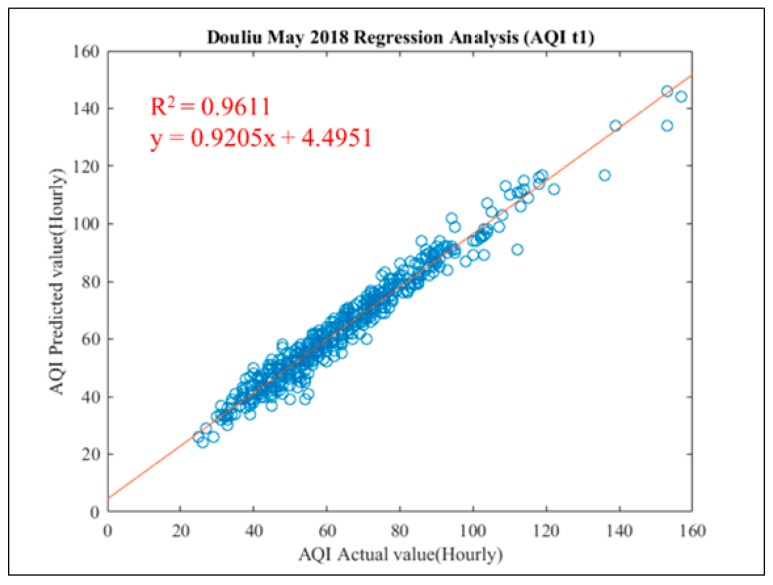
Regression analysis chart of AQI prediction in Douliu City (May 2018).

**Figure 14 ijerph-16-04679-f014:**
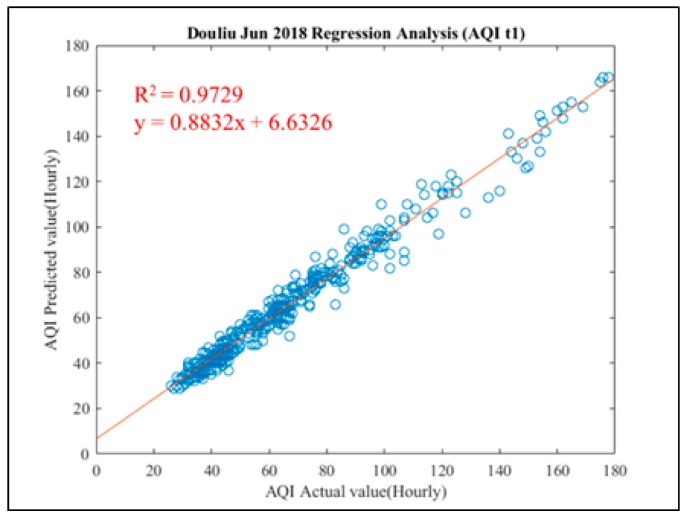
Regression analysis chart of AQI prediction in Douliu City (June 2018).

**Figure 15 ijerph-16-04679-f015:**
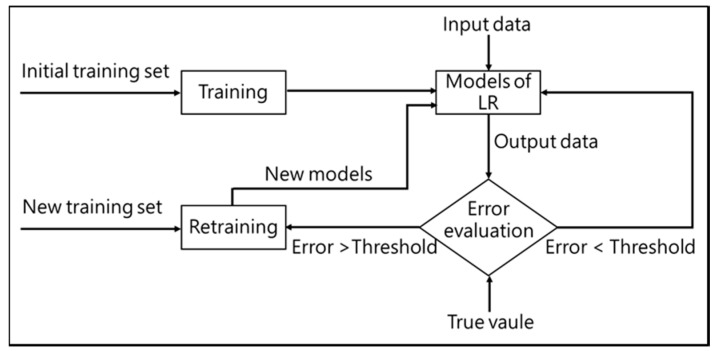
Model retraining process.

**Table 1 ijerph-16-04679-t001:** Table for individualized air quality index (IAQI) and AQI formulas.

Calculation	Symbol	Explanation
IAQI	IAQI_P_	Individual air quality index of pollutant item P.
	C_P_	Concentration value of pollutant item P.
	BP_Hi_	The upper limit for classification of pollutant items and CPs.
	BP_Lo_	The lower limit for classification of pollutant and CPs
	IAQI_Hi_	The upper limit of AQI classification corresponding to BPHi for pollutant items.
	IAQI_Lo_	The lower grading limit of AQI value corresponding to BPLo for pollutant items.
AQI	IAQI	Individual air quality index.
	n	Pollutant projects.

**Table 2 ijerph-16-04679-t002:** Research on Air Pollution in Recent Years.

Research Category	Method	Pollution Index	Author
Discussion on the Influences	Gauss Distribution	CO	Pan et al., [19]
	Statistical Analysis	PM_2.5_, NO_2_, SO_2_	Ng et al., [20]
		NO_2_, NO_x_	Hjortebjerg et al., [21]
		PM_10_, NO_2_, SO_2_	Deng et al., [22]
		PM_2.5_, PM_10_, NO_2_, SO_2_, CO, O_3_	Lee et al., [23]
		PM_10_, O_3_	Lichter et al., [24]
		PM_2.5_, BC	Kingsley et al., [25]
	Literature Review	PM_2.5_, PM_10_, NO_2_, SO_2_, CO, O_3_	Vizcaino et al., [26]
		PM_2.5_, PM_10_	Chen et al., [27]
			Santibáñez-Andrade et al., [28]
	Time Series	PM_10_, NO_2_, SO_2_	Ma et al., [29]
		PM_2.5_	Li et al., [30]
Prediction of Air Quality Index	Machine Learning	PM_2.5_	Perez & Gramsch, [31]
		NO_2_, SO_2_, O_3_	Shaban et al., [33]
		PM_2.5_	Zhan et al., [32]
		AQI	Wang et al., [33]
			Chen et al., [12]
	Statistical Model	PM_2.5_	Dong et al., [34]
			Xu & Wang, [35]
		AQI	Zhu et al., [11]
	Numerical Analysis	NO, NO_2_, SO_2_, CO, O_3_	Feng et al., [36]
	IoT Monitoring	PM_2.5_	Chen et al., [37]

**Table 3 ijerph-16-04679-t003:** Azure adopt function.

Item	Use of Efficiency Layer
App Service	B1 (Cores: 1, RAM: 1.75 GB, Storage: 10 GB, Disk Space: 10 GB)
SQL Database	S0 (DTUs: 10, Included Storage: 250 GB)
Machine Learning Studio	S1 (Included transactions: 100,000, Included compute hours: 25, Total number of web services: 10)
Storage	Standard

**Table 4 ijerph-16-04679-t004:** Testing results of different window size.

	Window Size	1	2	3	4
Performance					
MAE		3.833	3.868	3.824	3.633
RMSE		5.302	5.671	5.341	4.969
R^2^		0.881	0.868	0.882	0.898

**Table 5 ijerph-16-04679-t005:** Data Source Content and variables description.

Data Source	Variables	Data Field	Measurement/Units	Related Study
EPA	NA	Date Time	yyyy/MM/ddHH:mm:ss	Lee et al., [23] Vizcaino et al., [26] Wang et al., [3] Chen et al., [12] Zhu et al., [11]
	NA	Observatory Name	Station name/NA	
	*y*_1_(t + 1),…,*y*_1_(t + 6)	SO_2_(t + 1),…,SO_2_(t + 6)	Sulfur dioxide/ppb	
	*y*_2_(t + 1),…,*y*_2_(t + 6)	CO(t + 1),…,CO(t + 6)	Carbon monoxide/ppm	
	*y*_3_(t + 1),…,*y*_3_(t + 6)	O_3_(t + 1),…,O_3_(t + 6)	Ozone/ppb	
	*y*_4_(t + 1),…,*y*_4_(t + 6)	PM_10_(t + 1),…,PM_10_(t + 6)	Suspended particulates/μg/m^3^	
	*y*_5_(t + 1),…,*y*_5_(t + 6)	PM_2.5_(t + 1),…,PM_2.5_(t + 6)	Particulate matter/μg/m^3^	
	*y*_6_(t + 1),…,*y*_6_(t + 6)	NO_2_(t + 1),…,NO_2_(t + 6)	Nitrogen dioxide/ppb	
	*x_y1_*(t + 1),…,*x_y1_*(t + 5)	SO_2_(t + 1),…,SO_2_(t + 5)	Sulfur dioxide/ppb	
	*x_y2_*(t + 1),…,*x_y2_*(t + 5)	CO(t + 1),…,CO(t + 5)	Carbon monoxide/ppm	
	*x_y3_*(t + 1),…,*x_y3_*(t + 5)	O_3_(t + 1),…,O_3_(t + 5)	Ozone/ppb	
	*x_y4_*(t + 1),…,*x_y4_*(t + 5)	PM_10_(t + 1),…,PM_10_(t + 5)	Suspended particulates/μg/m^3^	
	*x_y5_*(t + 1),…,*x_y5_*(t + 5)	PM_2.5_(t + 1),…,PM_2.5_(t + 5)	Particulate matter/μg/m^3^	
	*x_y6_*(t + 1),…,*x_y6_*(t + 5)	NO_2_(t + 1),…,NO_2_(t + 5)	Nitrogen dioxide/ppb	
	*x*_1_(t − 3),…,*x*_1_(t)	SO_2_(t − 3),…,SO_2_(t)	Sulfur dioxide/ppb	
	*x*_2_(t − 3),…,*x*_2_(t)	CO(t − 3),…,CO(t)	Carbon monoxide/ppm	
	*x*_3_(t − 3),…,*x*_3_(t)	O_3_(t − 3),…,O_3_(t)	Ozone/ppb	
	*x*_4_(t − 3),…,*x*_4_(t)	PM_10_(t − 3),…,PM_10_(t)	Suspended particulates/μg/m^3^	
	*x*_5_(t − 3),…,*x*_5_(t)	PM_2.5_(t − 3),…,PM_2.5_(t)	Particulate matter/μg/m^3^	
	*x*_6_(t − 3),…,*x*_6_(t)	NO_2_(t − 3),…,NO_2_(t)	Nitrogen dioxide/ppb	
	*x*_7_(t − 3),…,*x*_7_(t)	NO_X_(t − 3),…,NO_X_(t)	Nitrogen oxide/ppb	Hjortebjerg et., [21]
	*x*_8_(t − 3),…,*x*_8_(t)	NO(t − 3),…,NO(t)	Nitric oxide/ppb	Feng et al., [36]
	*x*_9_(t − 3),…,*x*_9_(t)	AMB_TEMP(t − 3),…,AMB_TEMP(t)	Atmospheric temperature/°C	Voukantsis et al., [39] Sun et al., [40] Heyes et al., [41]
	*x*_10_(t − 3),…,*x*_10_(t)	RAINFALL(t − 3),…,RAINFALL(t)	Rainfall/mm	Sun et al., [40] Heyes et al., [41]
	*x*_11_(t − 3),…,*x*_11_(t)	RH(t − 3),…,RH(t)	Relative humidity/%	Voukantsis et al., [39] Sun et al., [40]
	*x*_12_(t − 3),…,*x*_12_(t)	WIND_SPEED(t − 3),…,WIND_SPEED(t)	Wind speed/m/sec	Heyes et al., [41]
	*x*_13_(t − 3),…,*x*_13_(t)	WIND_DIREC(t − 3),…,WIND_DIREC(t)	Wind direction/degress	
	*x*_14_(t − 3),…,*x*_14_(t)	WS_HR(t − 3),…,WS_HR(t)	Wind speed per hour/m/sec	Voukantsis et al., [39] Sun et al., [40]
	*x*_15_(t − 3),…,*x*_15_(t)	WD_HR(t − 3),…,WS_HR(t)	Wind direction per hour/degress	Heyes et al., [41] Li et al., [30]

Notes: *y_i_* = Output variables; *x_i_* = Input variables; *x_yi_* = Input Variables Predicted value of *y_i_*.

**Table 6 ijerph-16-04679-t006:** Comparison Table of AQI and classification.

AQI Value	Health Effects	Status in Color
0–50	good	green
51–100	ordinary	yellow
101–150	Poor to sensitive	orange
151–200	Bad	red
201–300	Very bad	purple
301–500	Harmful	maroon

**Table 7 ijerph-16-04679-t007:** Primitive Structure of Historical Data.

Data Field	Content
Date	(yyyy/MM/dd)
Station	Name of station (example: DouLiu, LunBei etc.)
Items	Monitoring items (example: SO_2_, CO, O_3_ etc.)
Hour	Hourly monitoring item values, 00~23 (24 h)

**Table 8 ijerph-16-04679-t008:** Real-time data original structure.

Data Field	Items/Unit	Notes
Observatory_Name	Station name/NA	Non-input variable
DateTime	yyyy/MM/dd HH:mm:ss	
SO_2_	Sulfur dioxide/ppb	
CO	Carbon monoxide/ppm	
O_3_	Ozone/ppb	
PM_10_	Suspended particulates/μg/m^3^	
PM_2.5_	Particulate matter/μg/m^3^	
NO_2_	Nitrogen dioxide/ppb	
NO_X_	Nitrogen oxide/ppb	
NO	Nitric oxide/ppb	
THC	Total hydrocarbon/ppm	Delete in subsequent processing
NMHC	Non-methane hydrocarbons/ppm	
CH_4_	Methane/ppm	
UVB	UV index/UVI	
AMB_TEMP	Atmospheric temperature/°C	
RAINFALL	Rainfall/mm	
RH	Relative humidity/%	
WIND_SPEED	Wind speed/m/sec	
WIND_DIREC	Wind direction/degress	
WS_HR	Wind speed per hour/m/sec	
WD_HR	Wind direction per hour/degress	
PH_RAIN	PH (acid rain)/pH	Delete in subsequent processing
RAIN_COND	Conductivity (acid rain)/μS/cm	

**Table 9 ijerph-16-04679-t009:** Performance comparison of algorithms and pollutant.

Pollutant	Algorithms	MAE	RMSE	R^2^
SO_2_	DFR	0.778	1.642	0.556
	LR	0.747	1.592	0.583
	NNR	0.793	1.624	0.566
CO	DFR	0.061	0.115	0.808
	LR	0.061	0.117	0.802
	NNR	0.059	0.112	0.817
O_3_	DFR	3.867	5.596	0.917
	LR	3.852	5.557	0.918
	NNR	3.967	5.611	0.916
PM_10_	DFR	5.144	7.758	0.926
	LR	4.849	7.452	0.932
	NNR	12.894	16.826	0.656
PM_2.5_	DFR	3.573	4.961	0.904
	LR	3.363	4.675	0.914
	NNR	4.784	6.201	0.850
NO_2_	DFR	2.539	3.756	0.839
	LR	2.461	3.641	0.848
	NNR	2.458	3.679	0.845

**Table 10 ijerph-16-04679-t010:** Overall Performance in Training Stage.

Pollutant	Performance	*Y*(t + 1)	*Y*(t + 2)	*Y*(t + 3)	*Y*(t + 4)	*Y*(t + 5)	*Y*(t + 6)
AQI	MAE	5.051	2.930	2.974	3.004	3.066	3.133
	RMSE	11.458	4.324	4.562	4.668	5.261	5.287
	R^2^	0.897	0.986	0.984	0.983	0.978	0.978
SO_2_	MAE	0.832	0.751	0.751	0.755	0.757	0.760
	RMSE	1.784	1.621	1.625	1.620	1.629	1.644
	R^2^	0.483	0.5740	0.569	0.571	0.568	0.562
CO	MAE	0.074	0.061	0.061	0.061	0.061	0.062
	RMSE	0.144	0.116	0.117	0.117	0.119	0.120
	R^2^	0.699	0.801	0.800	0.799	0.795	0.790
O_3_	MAE	5.235	3.909	3.953	3.976	4.008	4.091
	RMSE	9.335	5.683	5.787	5.849	5.964	6.160
	R^2^	0.773	0.913	0.910	0.908	0.905	0.899
PM_10_	MAE	6.263	4.861	4.875	4.877	4.948	4.955
	RMSE	11.071	7.609	7.644	7.641	7.977	7.860
	R^2^	0.853	0.929	0.928	0.928	0.922	0.924
PM_2.5_	MAE	4.134	3.373	3.388	3.390	3.422	3.440
	RMSE	6.453	4.747	4.789	4.802	4.951	4.921
	R^2^	0.839	0.912	0.911	0.910	0.905	0.906
NO_2_	MAE	3.000	2.479	2.482	2.478	2.478	2.500
	RMSE	4.942	3.678	3.695	3.687	3.714	3.781
	R^2^	0.725	0.844	0.842	0.842	0.839	0.833

**Table 11 ijerph-16-04679-t011:** Overall Performance in Test Stage.

Pollutant	Performance	*Y*(t + 1)	*Y*(t + 2)	*Y*(t + 3)	*Y*(t + 4)	*Y*(t + 5)	*Y*(t + 6)
AQI	MAE	3.124	6.001	8.649	12.843	12.069	13.420
	RMSE	4.516	8.319	11.762	18.080	15.984	17.613
	R^2^	0.981	0.936	0.870	0.683	0.758	0.705
SO_2_	MAE	0.674	0.921	1.046	1.196	1.184	1.223
	RMSE	1.277	1.600	1.744	1.922	1.894	1.934
	R^2^	0.635	0.426	0.316	0.174	0.196	0.161
CO	MAE	0.058	0.089	0.108	0.128	0.126	0.128
	RMSE	0.104	0.150	0.174	0.205	0.194	0.196
	R^2^	0.805	0.592	0.451	0.254	0.319	0.307
O_3_	MAE	3.765	6.268	8.227	11.182	11.183	12.223
	RMSE	5.310	8.490	10.930	14.958	14.569	15.821
	R^2^	0.922	0.801	0.671	0.395	0.418	0.316
PM_10_	MAE	5.107	8.382	10.832	14.046	13.473	14.389
	RMSE	7.746	12.428	15.813	20.445	19.210	20.758
	R^2^	0.926	0.810	0.393	0.489	0.544	0.490
PM_2.5_	MAE	3.456	5.010	6.209	7.654	7.423	7.889
	RMSE	4.785	6.954	8.625	10.576	10.204	10.773
	R^2^	0.896	7.781	0.664	0.502	0.531	0.479
NO_2_	MAE	2.556	3.848	4.658	5.560	5.521	5.690
	RMSE	3.759	5.392	6.356	7.628	7.328	7.505
	R^2^	0.841	0.671	0.541	0.344	0.384	0.352

**Table 12 ijerph-16-04679-t012:** Overall Performance for time t + 1 to t + 6.

Pollutant	Performance	*Y*(t + 1)	*Y*(t + 2)	*Y*(t + 3)	*Y*(t + 4)	*Y*(t + 5)	*Y*(t + 6)
AQI	MAE	3.246	5.936	8.076	9.283	10.430	11.242
	RMSE	5.983	10.110	13.426	15.140	16.466	17.262
	R^2^	0.947	0.853	0.728	0.638	0.555	0.506
SO_2_	MAE	0.766	1.026	1.146	1.221	1.269	1.301
	RMSE	1.442	1.787	1.932	2.026	2.084	2.111
	R^2^	0.592	0.380	0.282	0.217	0.171	0.146
CO	MAE	0.049	0.073	0.087	0.096	0.102	0.103
	RMSE	0.091	0.121	0.138	0.148	0.154	0.155
	R^2^	0.735	0.528	0.391	0.302	0.249	0.243
O_3_	MAE	1.239	6.750	8.758	10.375	12.013	13.136
	RMSE	6.857	9.611	12.035	13.946	16.103	17.554
	R^2^	0.895	0.795	0.682	0.579	0.422	0.344
PM_10_	MAE	4.513	6.870	8.380	9.070	9.775	10.265
	RMSE	7.451	10.927	12.856	13.797	14.748	15.267
	R^2^	0.852	0.674	0.530	0.444	0.361	0.131
PM_2.5_	MAE	2.923	3.925	4.565	4.792	5.116	5.350
	RMSE	3.991	5.276	6.108	6.368	6.785	7.036
	R^2^	0.767	0.601	0.470	0.421	0.351	0.314
NO_2_	MAE	2.072	2.966	3.520	3.917	4.200	4.369
	RMSE	3.046	4.059	4.661	5.099	5.425	5.603
	R^2^	0.786	0.622	0.503	0.406	0.328	0.281

**Table 13 ijerph-16-04679-t013:** Comparison of Studies.

	Shaban et al. [33]	Chen et al. [12]	Zhu et al. [11]	This Study
Computing platform	Local	Local	Local	Cloud
Prediction interval	1–24 h in the future	1 day in the future	1 h in the future	1–6 h in the future
Prediction target	SO_2_, O_3_, NO_2_	AQI, PM_2.5_, PHI, SSI	AQI	AQI, SO_2_, CO, O_3_, PM_10_, PM_2.5_, NO_2_
Research method	Machine Learning	Data Mining, Machine Learning	Machine Learning	Machine Learning
Algorithm	SVM, M5P, ANN	ANN	SVR	DFR, LR, NNR
Early Warning notice	N	N	N	Y
Visualization	N	N	N	Y

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
