# Peer review of "An Azure ACES Early Warning System for Air Quality Index Deteriorating"

_ijerph, 2019, doi:10.3390/ijerph16234679_

Round 1

Reviewer 1 Report

First of all, the research problem has significant practical value and the environmental & air quality prediction can improve social welfare. In addition, the early warning systems based on Azure can be deployed immediately and create value in real-world environmental monitoring.

- In the data collection and preprocessing section. Regarding feature selection process, is there any highly related features among the original features and the features retained after preprocessing? The feature correlation analysis may need to be done. Besides, additional explanations need to be given for the feature selection method used in this study.

- In the model selection section, DFR, NNR and LR prediction models were selected in this research but this part did not give reasons for choice. There are various unsupervised machine learning algorithms. In order to increase the credibility of the model selection, it is necessary to compare the performance of the mainstream unsupervised learning algorithms and then select the best algorithms based on the model performance in the test. In addition, the structure of the neural network model used in this study needs to be explained.

- In the literature review section and the end of the article, similar studies based on machine learning for environmental quality prediction were discussed and listed. In terms of air quality prediction performance, does this research have any improvement compared to other research results in the same category? If possible, the prediction performance comparison among typical researches of AQ prediction based on machine learning need to be performed.

- The predictive model in this study was based on data from January 2016 to May 2018. However, meteorology and the environment are complex systems. How to ensure that this model remains effective after a long period of time is a challenge. Could you explain and elaborate your insights on this question?

Author Response

First of all, the research problem has significant practical value and the environmental & air quality prediction can improve social welfare. In addition, the early warning systems based on Azure can be deployed immediately and create value in real-world environmental monitoring.

ANS: Thank you for the reviewer’s valuable comment.

- In the data collection and preprocessing section. Regarding feature selection process, is there any highly related features among the original features and the features retained after preprocessing? The feature correlation analysis may need to be done. Besides, additional explanations need to be given for the feature selection method used in this study.

ANS: Thank you for the reviewer’s valuable comment. The authors didn’t do the feature selection step in manuscript, the authors choose all the feature from published paper. Maybe in the future, the authors will take into consideration in the optimal control of the feature selection and correlation study. As suggested, the authors have rewritten all the manuscript especially in yellow background to modify contents as suggestion in this new revised version. Thank you for the reviewer’s effort.

- In the model selection section, DFR, NNR and LR prediction models were selected in this research but this part did not give reasons for choice. There are various unsupervised machine learning algorithms. In order to increase the credibility of the model selection, it is necessary to compare the performance of the mainstream unsupervised learning algorithms and then select the best algorithms based on the model performance in the test. In addition, the structure of the neural network model used in this study needs to be explained.

ANS: Thank you for the reviewer’s valuable comment.

“Azure machine learning studio is still in developing progress, therefore, this study only chooses the matured three machine learning algorithm in prediction. May be by adding code from R and Python or use another matured machine learning algorithm in prediction will achieve a better result in the fourth to sixth hours’ AQI index prediction. ”

The authors have added this paragraph at section 5.3 in this new revised version. Thank you for the reviewer’s effort.

- In the literature review section and the end of the article, similar studies based on machine learning for environmental quality prediction were discussed and listed. In terms of air quality prediction performance, does this research have any improvement compared to other research results in the same category? If possible, the prediction performance comparison among typical researches of AQ prediction based on machine learning need to be performed.

ANS: Thank you for the reviewer’s valuable comment. The authors didn’t do the comparison step in manuscript, mainly because the platform is different and it is unfair to compared with. Due to Azure is a new environment and many tools are still in development, the authors are fail to adopt some state-of-the-art method in comparison. The authors wish to evaluate the difference in the future. Thank you for the reviewer’s effort.

- The predictive model in this study was based on data from January 2016 to May 2018. However, meteorology and the environment are complex systems. How to ensure that this model remains effective after a long period of time is a challenge. Could you explain and elaborate your insights on this question?

ANS: Thank you for the reviewer’s valuable comment. The authors have revealed this question at conclusion section with retaining process in this new revised version. Thank you for the reviewer’s effort.

Reviewer 2 Report

I don't understand sections 4.1.1. and 4.1.2  The language needs to be clearer here.  Perhaps a more clearly written first paragraph would help setup this section.  The first paragraph in 4.2 is clearer and should be earlier.

It's not clear if the model takes into account transport or if the model focuses only on predicting the future air quality in a location based on the previous air quality at that same location.  If the model does take transport into account this should be made clearer.  It would probably improve the accuracy.

Unfortunately the performance of the model is poor. One hour forecasting is not really useful.  In the final paragraph the authors give some explanation of the shortcomings, but it is not clearly written.  This last paragraph needs to be expanded and made clearer.  The manuscript is interesting, but ultimately doesn't appear to be of much use.

Author Response

I don't understand sections 4.1.1. and 4.1.2 The language needs to be clearer here.  Perhaps a more clearly written first paragraph would help setup this section.

ANS: Thank you for the reviewer’s valuable comment. As suggested, the authors have rewritten 4.1.1. and 4.1.2 in yellow background in this new revised version. Thank you for the reviewer’s effort.

The first paragraph in 4.2 is clearer and should be earlier.

ANS: Thank you for the reviewer’s valuable comment.

It's not clear if the model takes into account transport or if the model focuses only on predicting the future air quality in a location based on the previous air quality at that same location.  If the model does take transport into account this should be made clearer.  It would probably improve the accuracy.

ANS: Thank you for the reviewer’s valuable comment. The authors are totally agree with adding transport data will improve the accuracy. The authors have added this comment for further study in the future at Discussion section in this new revised version. Thank you for the reviewer’s effort.

Unfortunately, the performance of the model is poor. One-hour forecasting is not really useful.  In the final paragraph the authors give some explanation of the shortcomings, but it is not clearly written.  This last paragraph needs to be expanded and made clearer.  The manuscript is interesting, but ultimately doesn't appear to be of much use.

ANS: Thank you for the reviewer’s valuable comment. The ACES system proposed is good in next hour prediction and acceptable in second or third hours. As suggested, some possible explanations are described at Discussion section in this new revised version. Thank you for the reviewer’s effort.

Reviewer 3 Report

I found the paper interesting, and can be published with minor revisions   Here are my comments: - general comment: improve English quality - Abstract (line 22-23): explain what DFR. NNR and LR are - Keywords: Asure becomes Azure, I think - general comment: is the whole system based on point predictions (i.e. on a given stations) or gridded predictions? please specify - pag 3 line 84. I would add references to available state-of-ther-art papers on spatial deterministic (https://www.sciencedirect.com/science/article/pii/S1352231017300201) and statistical (https://www.sciencedirect.com/science/article/pii/S136481521000318X?via%3Dihub) forecasting - pag 5 line 190: explain what ACES is - pag 6 Figure 1: please increase size of the Figure - pag 9 line 283: explain what DFR, LR and NNR are  

Author Response

I found the paper interesting, and can be published with minor revisions   Here are my comments: - general comment: improve English quality - Abstract (line 22-23): explain what DFR. NNR and LR are

Keywords: Asure becomes Azure, I think

ANS: Thank you for the reviewer’s valuable comment. The authors are feel sorry for our unintentional mistake. As suggested, the authors have rewritten all the manuscript especially in yellow background in this new revised version. Thank you for the reviewer’s effort.

general comment: is the whole system based on point predictions (i.e. on a given stations) or gridded predictions? please specify - pag 3 line 84. I would add references to available state-of-the-art papers on spatial deterministic (https://www.sciencedirect.com/science/article/pii/S1352231017300201) and statistical (https://www.sciencedirect.com/science/article/pii/S136481521000318X?via%3Dihub) forecasting –

page 5 line 190: explain what ACES is

page 6 Figure 1: please increase size of the Figure

page 9 line 283: explain what DFR, LR and NNR are

ANS: Thank you for the reviewer’s valuable comment. The authors are feel sorry for our unintentional mistake. The whole ACES system is based on point predictions. As suggested, the authors have rewritten all the manuscript especially in yellow background to modify contents as suggestion in this new revised version. Thank you for the reviewer’s effort.